# Non-strongly-convex smooth stochastic approximation with convergence rate $O(1/n)$

**Francis Bach**
INRIA - Sierra Project-team
Ecole Normale Supérieure, Paris, France
francis.bach@ens.fr

**Eric Moulines**
LTCI
Telecom ParisTech, Paris, France
eric.moulines@enst.fr

## Abstract

We consider the stochastic approximation problem where a convex function has to be minimized, given only the knowledge of unbiased estimates of its gradients at certain points, a framework which includes machine learning methods based on the minimization of the empirical risk. We focus on problems without strong convexity, for which all previously known algorithms achieve a convergence rate for function values of $O(1/\sqrt{n})$ after $n$ iterations. We consider and analyze two algorithms that achieve a rate of $O(1/n)$ for classical supervised learning problems. For least-squares regression, we show that *averaged* stochastic gradient descent *with constant step-size* achieves the desired rate. For logistic regression, this is achieved by a simple novel stochastic gradient algorithm that (a) constructs successive local quadratic approximations of the loss functions, while (b) preserving the same running-time complexity as stochastic gradient descent. For these algorithms, we provide a non-asymptotic analysis of the generalization error (in expectation, and also in high probability for least-squares), and run extensive experiments showing that they often outperform existing approaches.

## 1 Introduction

Large-scale machine learning problems are becoming ubiquitous in many areas of science and engineering. Faced with large amounts of data, practitioners typically prefer algorithms that process each observation only once, or a few times. Stochastic approximation algorithms such as stochastic gradient descent (SGD) and its variants, although introduced more than sixty years ago [1], still remain the most widely used and studied method in this context (see, e.g., [2, 3, 4, 5, 6, 7]).

We consider minimizing convex functions $f$, defined on a Euclidean space $\mathcal{F}$, given by $f(\theta) = \mathbb{E}\big[\ell(y, \langle\theta, x\rangle)\big]$, where $(x, y) \in \mathcal{F} \times \mathbb{R}$ denotes the data and $\ell$ denotes a loss function that is convex with respect to the second variable. This includes logistic and least-squares regression. In the stochastic approximation framework, independent and identically distributed pairs $(x_n, y_n)$ are observed sequentially and the predictor defined by $\theta$ is updated after each pair is seen.

We partially understand the properties of $f$ that affect the problem difficulty. *Strong convexity* (i.e., when $f$ is twice differentiable, a uniform strictly positive lower-bound $\mu$ on Hessians of $f$) is a key property. Indeed, after $n$ observations and with the proper step-sizes, averaged SGD achieves the rate of $O(1/\mu n)$ in the strongly-convex case [5, 4], while it achieves only $O(1/\sqrt{n})$ in the non-strongly-convex case [5], with matching lower-bounds [8].

The main issue with strong convexity is that typical machine learning problems are high-dimensional and have correlated variables so that the strong convexity constant $\mu$ is zero or very close to zero, and in any case smaller than $O(1/\sqrt{n})$. This then makes the non-strongly convex methods better. In this paper, we aim at obtaining algorithms that may deal with arbitrarily small strong-convexity constants, but still achieve a rate of $O(1/n)$.

*Smoothness* plays a central role in the context of deterministic optimization. The known convergence rates for smooth optimization are better than for non-smooth optimization (e.g., see [9]). However, for stochastic optimization the use of smoothness only leads to improvements on constants (e.g., see [10]) but not on the rate itself, which remains $O(1/\sqrt{n})$ for non-strongly-convex problems.

We show that for the square loss and for the logistic loss, we may use the smoothness of the loss and obtain algorithms that have a convergence rate of $O(1/n)$ without any strong convexity assumptions. More precisely, for least-squares regression, we show in Section 2 that *averaged* stochastic gradient descent *with constant step-size* achieves the desired rate. For logistic regression this is achieved by a novel stochastic gradient algorithm that (a) constructs successive local quadratic approximations of the loss functions, while (b) preserving the same running-time complexity as stochastic gradient descent (see Section 3). For these algorithms, we provide a non-asymptotic analysis of their generalization error (in expectation, and also in high probability for least-squares), and run extensive experiments on standard machine learning benchmarks showing in Section 4 that they often outperform existing approaches.

## 2 Constant-step-size least-mean-square algorithm

In this section, we consider stochastic approximation for least-squares regression, where SGD is often referred to as the least-mean-square (LMS) algorithm. The novelty of our convergence result is the use of the constant step-size with averaging, which was already considered by [11], but now with an explicit non-asymptotic rate $O(1/n)$ without any dependence on the lowest eigenvalue of the covariance matrix.

### 2.1 Convergence in expectation

We make the following assumptions:

**(A1)** $\mathcal{F}$ is a $d$-dimensional Euclidean space, with $d \geqslant 1$.

**(A2)** The observations $(x_n, z_n) \in \mathcal{F} \times \mathcal{F}$ are independent and identically distributed.

**(A3)** $\mathbb{E}\|x_n\|^2$ and $\mathbb{E}\|z_n\|^2$ are finite. Denote by $H = \mathbb{E}(x_n \otimes x_n)$ the covariance operator from $\mathcal{F}$ to $\mathcal{F}$. Without loss of generality, $H$ is assumed invertible (by projecting onto the minimal subspace where $x_n$ lies almost surely). However, its eigenvalues may be arbitrarily small.

**(A4)** The global minimum of $f(\theta) = (1/2)\mathbb{E}\big[\langle \theta, x_n\rangle^2 - 2\langle \theta, z_n\rangle\big]$ is attained at a certain $\theta_* \in \mathcal{F}$. We denote by $\xi_n = z_n - \langle \theta_*, x_n\rangle x_n$ the residual. We have $\mathbb{E}\big[\xi_n\big] = 0$, but in general, it is not true that $\mathbb{E}\big[\xi_n \mid x_n\big] = 0$ (unless the model is well-specified).

**(A5)** We study the stochastic gradient (a.k.a. least mean square) recursion defined as

$$\theta_n = \theta_{n-1} - \gamma(\langle \theta_{n-1}, x_n\rangle x_n - z_n) = (I - \gamma x_n \otimes x_n)\theta_{n-1} + \gamma z_n, \tag{1}$$

started from $\theta_0 \in \mathcal{F}$. We also consider the averaged iterates $\bar{\theta}_n = (n+1)^{-1}\sum_{k=0}^{n}\theta_k$.

**(A6)** There exists $R > 0$ and $\sigma > 0$ such that $\mathbb{E}\big[\xi_n \otimes \xi_n\big] \preccurlyeq \sigma^2 H$ and $\mathbb{E}\big(\|x_n\|^2 x_n \otimes x_n\big) \preccurlyeq R^2 H$, where $\preccurlyeq$ denotes the the order between self-adjoint operators, i.e., $A \preccurlyeq B$ if and only if $B - A$ is positive semi-definite.

**Discussion of assumptions.** Assumptions **(A1-5)** are standard in stochastic approximation (see, e.g., [12, 6]). Note that for least-squares problems, $z_n$ is of the form $y_n x_n$, where $y_n \in \mathbb{R}$ is the response to be predicted as a linear function of $x_n$. We consider a slightly more general case than least-squares because we will need it for the quadratic approximation of the logistic loss in Section 3.1. Note that in assumption **(A4)**, we do not assume that the model is well-specified.

Assumption **(A6)** is true for least-square regression with almost surely bounded data, since, if $\|x_n\|^2 \leqslant R^2$ almost surely, then $\mathbb{E}\big(\|x_n\|^2 x_n \otimes x_n\big) \preccurlyeq \mathbb{E}\big(R^2 x_n \otimes x_n\big) = R^2 H$; a similar inequality holds for the output variables $y_n$. Moreover, it also holds for data with infinite supports, such as Gaussians or mixtures of Gaussians (where all covariance matrices of the mixture components are lower and upper bounded by a constant times the same matrix). Note that the finite-dimensionality assumption could be relaxed, but this would require notions similar to degrees of freedom [13], which is outside of the scope of this paper.

The goal of this section is to provide a non-asymptotic bound on the expectation $\mathbb{E}\big[f(\bar{\theta}_n) - f(\theta_*)\big]$, that (a) does not depend on the smallest non-zero eigenvalue of $H$ (which could be arbitrarily small) and (b) still scales as $O(1/n)$.

**Theorem 1** *Assume (A1-6). For any constant step-size $\gamma < 1/R^2$, we have*

$$\mathbb{E}\big[f(\bar{\theta}_{n-1}) - f(\theta_*)\big] \leqslant \frac{1}{2n}\left[\frac{\sigma\sqrt{d}}{1 - \sqrt{\gamma R^2}} + R\|\theta_0 - \theta_*\|\frac{1}{\sqrt{\gamma R^2}}\right]^2. \tag{2}$$

*When $\gamma = 1/(4R^2)$, we obtain $\mathbb{E}\big[f(\bar{\theta}_{n-1}) - f(\theta_*)\big] \leqslant \frac{2}{n}\big[\sigma\sqrt{d} + R\|\theta_0 - \theta_*\|\big]^2.$*

**Proof technique.** We adapt and extend a proof technique from [14] which is based on non-asymptotic expansions in powers of $\gamma$. We also use a result from [2] which studied the recursion in Eq. (1), with $x_n \otimes x_n$ replaced by its expectation $H$. See [15] for details.

**Optimality of bounds.** Our bound in Eq. (2) leads to a rate of $O(1/n)$, which is known to be optimal for least-squares regression (i.e., under reasonable assumptions, no algorithm, even more complex than averaged SGD can have a better dependence in $n$) [16]. The term $\sigma^2 d/n$ is also unimprovable.

**Initial conditions.** If $\gamma$ is small, then the initial condition is forgotten more slowly. Note that with additional strong convexity assumptions, the initial condition would be forgotten faster (exponentially fast without averaging), which is one of the traditional uses of constant-step-size LMS [17].

**Specificity of constant step-sizes.** The non-averaged iterate sequence $(\theta_n)$ is a homogeneous Markov chain; under appropriate technical conditions, this Markov chain has a unique stationary (invariant) distribution and the sequence of iterates $(\theta_n)$ converges in distribution to this invariant distribution; see [18, Chapter 17]. Denote by $\pi_\gamma$ the invariant distribution. Assuming that the Markov Chain is Harris-recurrent, the ergodic theorem for Harris Markov chain shows that $\bar{\theta}_{n-1} = n^{-1}\sum_{k=0}^{n-1}\theta_k$ converges almost-surely to $\bar{\theta}_\gamma \stackrel{\text{def}}{=} \int \theta\pi_\gamma(\mathrm{d}\theta)$, which is the mean of the stationary distribution. Taking the expectation on both side of Eq. (1), we get $\mathbb{E}[\theta_n] - \theta_* = (I - \gamma H)(\mathbb{E}[\theta_{n-1}] - \theta_*)$, which shows, using that $\lim_{n\to\infty}\mathbb{E}[\theta_n] = \bar{\theta}_\gamma$ that $H\bar{\theta}_\gamma = H\theta_*$ and therefore $\bar{\theta}_\gamma = \theta_*$ since $H$ is invertible. Under slightly stronger assumptions, it can be shown that

$$\lim_{n\to\infty} n\mathbb{E}[(\bar{\theta}_n - \theta_*)^2] = \mathrm{Var}_{\pi_\gamma}(\theta_0) + 2\sum_{k=1}^{\infty}\mathrm{Cov}_{\pi_\gamma}(\theta_0, \theta_k),$$

where $\mathrm{Cov}_{\pi_\gamma}(\theta_0, \theta_k)$ denotes the covariance of $\theta_0$ and $\theta_k$ when the Markov chain is started from stationarity. This implies that $\lim_{n\to\infty} n\mathbb{E}[f(\bar{\theta}_n) - f(\theta_*)]$ has a finite limit. Therefore, this interpretation explains why the averaging produces a sequence of estimators which converges to the solution $\theta_*$ pointwise, and that the rate of convergence of $\mathbb{E}[f(\theta_n) - f(\theta_*)]$ is of order $O(1/n)$. Note that (a) our result is stronger since it is independent of the lowest eigenvalue of $H$, and (b) for other losses than quadratic, the same properties hold *except* that the mean under the stationary distribution does not coincide with $\theta_*$ and its distance to $\theta_*$ is typically of order $\gamma^2$ (see Section 3).

## 2.2 Convergence in higher orders

We are now going to consider an extra assumption in order to bound the $p$-th moment of the excess risk and then get a high-probability bound. Let $p$ be a real number greater than 1.

**(A7)** There exists $R > 0$, $\kappa > 0$ and $\tau \geqslant \sigma > 0$ such that, for all $n \geqslant 1$, $\|x_n\|^2 \leqslant R^2$ a.s., and

$$\mathbb{E}\|\xi_n\|^p \leqslant \tau^p R^p \quad \text{and} \quad \mathbb{E}\big[\xi_n \otimes \xi_n\big] \preccurlyeq \sigma^2 H, \tag{3}$$

$$\forall z \in \mathcal{F}, \quad \mathbb{E}\langle z, x_n\rangle^4 \leqslant \kappa\big(\mathbb{E}\langle z, x_n\rangle^2\big)^2 = \kappa\langle z, Hz\rangle^2. \tag{4}$$

The last condition in Eq. (4) says that the *kurtosis* of the projection of the covariates $x_n$ on any direction $z \in \mathcal{F}$ is bounded. Note that computing the constant $\kappa$ happens to be equivalent to the optimization problem solved by the FastICA algorithm [19], which thus provides an estimate of $\kappa$. In Table 1, we provide such an estimate for the non-sparse datasets which we have used in experiments, while we consider only directions $z$ along the axes for high-dimensional sparse datasets. For these datasets where a given variable is equal to zero except for a few observations, $\kappa$ is typically quite large. Adapting and analyzing normalized LMS techniques [20] to this set-up is likely to improve the theoretical robustness of the algorithm (but note that results in expectation from Theorem 1 do not use $\kappa$). The next theorem provides a bound for the $p$-th moment of the excess risk.

**Theorem 2** *Assume (A1-7). For any real $p \geqslant 1$, and for a step-size $\gamma \leqslant 1/(12p\kappa R^2)$, we have:*

$$\big(\mathbb{E}\big|f(\bar{\theta}_{n-1}) - f(\theta_*)\big|^p\big)^{1/p} \leqslant \frac{p}{2n}\left(7\tau\sqrt{d} + R\|\theta_0 - \theta_*\|\sqrt{3 + \frac{2}{\gamma p R^2}}\right)^2. \tag{5}$$

*For $\gamma = 1/(12p\kappa R^2)$, we get:* $\left(\mathbb{E}\big|f(\bar{\theta}_{n-1}) - f(\theta_*)\big|^p\right)^{1/p} \leqslant \frac{p}{2n}\left(7\tau\sqrt{d} + 6\sqrt{\kappa}R\|\theta_0 - \theta_*\|\right)^2.$

Note that to control the $p$-th order moment, a smaller step-size is needed, which scales as $1/p$. We can now provide a high-probability bound; the tails decay polynomially as $1/(n\delta^{12\gamma\kappa R^2})$ and the smaller the step-size $\gamma$, the lighter the tails.

**Corollary 1** *For any step-size such that $\gamma \leqslant 1/(12\kappa R^2)$, any $\delta \in (0,1)$,*

$$\mathbb{P}\left(f(\bar{\theta}_{n-1}) - f(\theta_*) \geqslant \frac{1}{n\delta^{12\gamma\kappa R^2}} \frac{\left[7\tau\sqrt{d} + R\|\theta_0 - \theta_*\|(\sqrt{3} + \sqrt{24\kappa})\right]^2}{24\gamma\kappa R^2}\right) \leqslant \delta. \tag{6}$$

## 3   Beyond least-squares: M-estimation

In Section 2, we have shown that for least-squares regression, averaged SGD achieves a convergence rate of $O(1/n)$ with no assumption regarding strong convexity. For all losses, with a constant step-size $\gamma$, the stationary distribution $\pi_\gamma$ corresponding to the homogeneous Markov chain $(\theta_n)$ does always satisfy $\int f'(\theta)\pi_\gamma(\mathrm{d}\theta) = 0$, where $f$ is the generalization error. When the gradient $f'$ is linear (i.e., $f$ is quadratic), then this implies that $f'(\int \theta\pi_\gamma(\mathrm{d}\theta)) = 0$, i.e., the averaged recursion converges pathwise to $\bar{\theta}_\gamma = \int \theta\pi_\gamma(\mathrm{d}\theta)$ which coincides with the optimal value $\theta_*$ (defined through $f'(\theta_*) = 0$). When the gradient $f'$ is no longer linear, then $\int f'(\theta)\pi_\gamma(\mathrm{d}\theta) \neq f'(\int \theta\pi_\gamma(\mathrm{d}\theta))$. Therefore, for general $M$-estimation problems we should expect that the averaged sequence still converges at rate $O(1/n)$ to the mean of the stationary distribution $\bar{\theta}_\gamma$, but not to the optimal predictor $\theta_*$. Typically, the average distance between $\theta_n$ and $\theta_*$ is of order $\gamma$ (see Section 4 and [21]), while for the averaged iterates that converge pointwise to $\bar{\theta}_\gamma$, it is of order $\gamma^2$ for strongly convex problems under some additional smoothness conditions on the loss functions (these are satisfied, for example, by the logistic loss [22]).

Since quadratic functions may be optimized with rate $O(1/n)$ under weak conditions, we are going to use a quadratic approximation around a well chosen support point, which shares some similarity with the Newton procedure (however, with a non trivial adaptation to the stochastic approximation framework). The Newton step for $f$ around a certain point $\tilde{\theta}$ is equivalent to minimizing a quadratic surrogate $g$ of $f$ around $\tilde{\theta}$, i.e., $g(\theta) = f(\tilde{\theta}) + \langle f'(\tilde{\theta}), \theta - \tilde{\theta}\rangle + \frac{1}{2}\langle\theta - \tilde{\theta}, f''(\tilde{\theta})(\theta - \tilde{\theta})\rangle$. If $f_n(\theta) \overset{\text{def}}{=} \ell(y_n, \langle\theta, x_n\rangle)$, then $g(\theta) = \mathbb{E}g_n(\theta)$, with $g_n(\theta) = f(\tilde{\theta}) + \langle f'_n(\tilde{\theta}), \theta - \tilde{\theta}\rangle + \frac{1}{2}\langle\theta - \tilde{\theta}, f''_n(\tilde{\theta})(\theta - \tilde{\theta})\rangle$; the Newton step may thus be solved approximately with stochastic approximation (here constant-step size LMS), with the following recursion:

$$\theta_n = \theta_{n-1} - \gamma g'_n(\theta_{n-1}) = \theta_{n-1} - \gamma\left[f'_n(\tilde{\theta}) + f''_n(\tilde{\theta})(\theta_{n-1} - \tilde{\theta})\right]. \tag{7}$$

This is equivalent to replacing the gradient $f'_n(\theta_{n-1})$ by its first-order approximation around $\tilde{\theta}$. A crucial point is that for machine learning scenarios where $f_n$ is a loss associated to a single data point, its complexity is only twice the complexity of a regular stochastic approximation step, since, with $f_n(\theta) = \ell(y_n, \langle x_n, \theta\rangle)$, $f''_n(\theta)$ is a rank-one matrix.

**Choice of support points for quadratic approximation.** An important aspect is the choice of the support point $\tilde{\theta}$. In this paper, we consider two strategies:

– **Two-step procedure**: for convex losses, averaged SGD with a step-size decaying at $O(1/\sqrt{n})$ achieves a rate (up to logarithmic terms) of $O(1/\sqrt{n})$ [5, 6]. We may thus use it to obtain a first decent estimate. The two-stage procedure is as follows (and uses $2n$ observations): $n$ steps of averaged SGD with constant step size $\gamma \propto 1/\sqrt{n}$ to obtain $\tilde{\theta}$, and then averaged LMS for the Newton step around $\tilde{\theta}$. As shown below, this algorithm achieves the rate $O(1/n)$ for logistic regression. However, it is not the most efficient in practice.

– **Support point = current average iterate**: we simply consider the current averaged iterate $\bar{\theta}_{n-1}$ as the support point $\tilde{\theta}$, leading to the recursion:

$$\theta_n = \theta_{n-1} - \gamma\left[f'_n(\bar{\theta}_{n-1}) + f''_n(\bar{\theta}_{n-1})(\theta_{n-1} - \bar{\theta}_{n-1})\right]. \tag{8}$$

Although this algorithm has shown to be the most efficient in practice (see Section 4) we currently have no proof of convergence. Given that the behavior of the algorithms does not change much when the support point is updated less frequently than each iteration, there may be some connections to two-time-scale algorithms (see, e.g., [23]). In Section 4, we also consider several other strategies based on doubling tricks.

Interestingly, for non-quadratic functions, our algorithm imposes a new bias (by replacing the true gradient by an approximation which is only valid close to $\bar{\theta}_{n-1}$) in order to reach faster convergence (due to the linearity of the underlying gradients).

**Relationship with one-step-estimators.** One-step estimators (see, e.g., [24]) typically take any estimator with $O(1/n)$-convergence rate, and make a full Newton step to obtain an efficient estimator (i.e., one that achieves the Cramer-Rao lower bound). Although our novel algorithm is largely inspired by one-step estimators, our situation is slightly different since our first estimator has only convergence rate $O(1/\sqrt{n})$ and is estimated on different observations.

### 3.1 Self-concordance and logistic regression

We make the following assumptions:

**(B1)** $\mathcal{F}$ is a $d$-dimensional Euclidean space, with $d \geqslant 1$.

**(B2)** The observations $(x_n, y_n) \in \mathcal{F} \times \{-1, 1\}$ are independent and identically distributed.

**(B3)** We consider $f(\theta) = \mathbb{E}\big[\ell(y_n, \langle x_n, \theta \rangle)\big]$, with the following assumption on the loss function $\ell$ (whenever we take derivatives of $\ell$, this will be with respect to the second variable):

$$\forall (y, \hat{y}) \in \{-1, 1\} \times \mathbb{R}, \quad \ell'(y, \hat{y}) \leqslant 1, \quad \ell''(y, \hat{y}) \leqslant 1/4, \quad |\ell'''(y, \hat{y})| \leqslant \ell''(y, \hat{y}).$$

We denote by $\theta_*$ a global minimizer of $f$, which we thus assume to exist, and we denote by $H = f''(\theta_*)$ the Hessian operator at a global optimum $\theta_*$.

**(B4)** We assume that there exists $R > 0$, $\kappa > 0$ and $\rho > 0$ such that $\|x_n\|^2 \leqslant R^2$ almost surely, and

$$\mathbb{E}\big[x_n \otimes x_n\big] \preccurlyeq \rho \mathbb{E}\big[\ell''(y_n, \langle \theta_*, x_n \rangle) x_n \otimes x_n\big] = \rho H, \tag{9}$$

$$\forall z \in \mathcal{F}, \theta \in \mathcal{F}, \ \mathbb{E}\big[\ell''(y_n, \langle \theta, x_n \rangle)^2 \langle z, x_n \rangle^4\big] \leqslant \kappa\big(\mathbb{E}\big[\ell''(y_n, \langle \theta, x_n \rangle) \langle z, x_n \rangle^2\big]\big)^2. \tag{10}$$

Assumption **(B3)** is satisfied for the logistic loss and extends to all generalized linear models (see more details in [22]), and the relationship between the third derivative and second derivative of the loss $\ell$ is often referred to as *self-concordance* (see [9, 25] and references therein). Note moreover that we must have $\rho \geqslant 4$ and $\kappa \geqslant 1$.

A loose upper bound for $\rho$ is $1/\inf_n \ell''(y_n, \langle \theta_*, x_n \rangle)$ but in practice, it is typically much smaller (see Table 1). The condition in Eq. (10) is hard to check because it is uniform in $\theta$. With a slightly more complex proof, we could restrict $\theta$ to be close to $\theta_*$; with such constraints, the value of $\kappa$ we have found is close to the one from Section 2.2 (i.e., without the terms in $\ell''(y_n, \langle \theta, x_n \rangle)$).

**Theorem 3** *Assume (B1-4), and consider the vector $\zeta_n$ obtained as follows: (a) perform $n$ steps of averaged stochastic gradient descent with constant step size $1/2R^2\sqrt{n}$, to get $\tilde{\theta}_n$, and (b) perform $n$ step of averaged LMS with constant step-size $1/R^2$ for the quadratic approximation of $f$ around $\tilde{\theta}_n$. If $n \geqslant (19 + 9R\|\theta_0 - \theta_*\|)^4$, then*

$$\mathbb{E}f(\zeta_n) - f(\theta_*) \leqslant \frac{\kappa^{3/2}\rho^3 d}{n}(16R\|\theta_0 - \theta_*\| + 19)^4. \tag{11}$$

We get an $O(1/n)$ convergence rate without assuming strong convexity, even locally, thus improving on results from [22] where the the rate is proportional to $1/(n\lambda_{\min}(H))$. The proof relies on self-concordance properties and the sharp analysis of the Newton step (see [15] for details).

## 4 Experiments

### 4.1 Synthetic data

**Least-mean-square algorithm.** We consider normally distributed inputs, with covariance matrix $H$ that has random eigenvectors and eigenvalues $1/k$, $k = 1, \ldots, d$. The outputs are generated from a linear function with homoscedastic noise with unit signal to noise-ratio. We consider $d = 20$ and the least-mean-square algorithm with several settings of the step size $\gamma_n$, constant or proportional to $1/\sqrt{n}$. Here $R^2$ denotes the *average radius of the data*, i.e., $R^2 = \operatorname{tr} H$. In the left plot of Figure 1, we show the results, averaged over 10 replications.

Without averaging, the algorithm with constant step-size does not converge pointwise (it oscillates), and its average excess risk decays as a linear function of $\gamma$ (indeed, the gap between each values of the constant step-size is close to $\log_{10}(4)$, which corresponds to a linear function in $\gamma$).

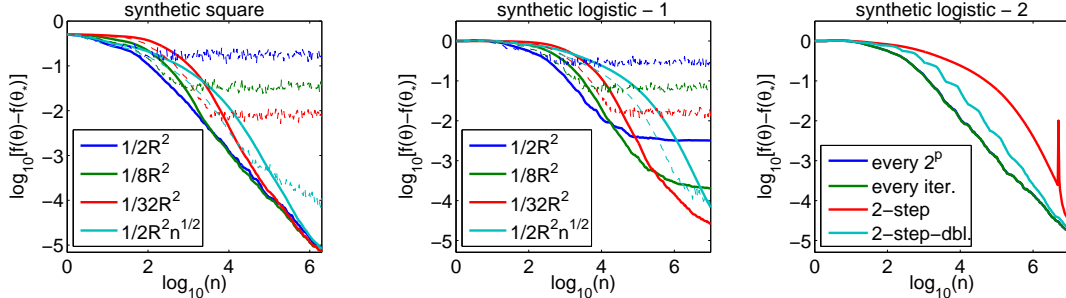

Figure 1: Synthetic data. Left: least-squares regression. Middle: logistic regression with averaged SGD with various step-sizes, averaged (plain) and non-averaged (dashed). Right: various Newton-based schemes for the same logistic regression problem. Best seen in color; see text for details.

With averaging, the algorithm with constant step-size does converge at rate $O(1/n)$, and for all values of the constant $\gamma$, the rate is actually the same. Moreover (although it is not shown in the plots), the standard deviation is much lower.

With decaying step-size $\gamma_n = 1/(2R^2\sqrt{n})$ and without averaging, the convergence rate is $O(1/\sqrt{n})$, and improves to $O(1/n)$ with averaging.

**Logistic regression.** We consider the same input data as for least-squares, but now generates outputs from the logistic probabilistic model. We compare several algorithms and display the results in Figure 1 (middle and right plots).

On the middle plot, we consider SGD; without averaging, the algorithm with constant step-size does not converge and its average excess risk reaches a constant value which is a linear function of $\gamma$ (indeed, the gap between each values of the constant step-size is close to $\log_{10}(4)$). With averaging, the algorithm does converge, but as opposed to least-squares, to a point which is not the optimal solution, with an error proportional to $\gamma^2$ (the gap between curves is twice as large).

On the right plot, we consider various variations of our online Newton-approximation scheme. The "2-step" algorithm is the one for which our convergence rate holds ($n$ being the total number of examples, we perform $n/2$ steps of averaged SGD, then $n/2$ steps of LMS). Not surprisingly, it is not the best in practice (in particular at $n/2$, when starting the constant-size LMS, the performance worsens temporarily). It is classical to use doubling tricks to remedy this problem while preserving convergence rates [26], this is done in "2-step-dbl.", which avoids the previous erratic behavior.

We have also considered getting rid of the first stage where plain averaged stochastic gradient is used to obtain a support point for the quadratic approximation. We now consider only Newton-steps but change only these support points. We consider updating the support point at every iteration, i.e., the recursion from Eq. (8), while we also consider updating it every dyadic point ("dbl.-approx"). The last two algorithms perform very similarly and achieve the $O(1/n)$ early. In all experiments on real data, we have considered the simplest variant (which corresponds to Eq. (8)).

## 4.2   Standard benchmarks

We have considered 6 benchmark datasets which are often used in comparing large-scale optimization methods. The datasets are described in Table 1 and vary in values of $d$, $n$ and sparsity levels. These are all *finite* binary classification datasets with outputs in $\{-1, 1\}$. For least-squares and logistic regression, we have followed the following experimental protocol: (1) remove all outliers (i.e., sample points $x_n$ whose norm is greater than 5 times the average norm), (2) divide the dataset in two equal parts, one for training, one for testing, (3) sample within the training dataset with replacement, for 100 times the number of observations in the training set (this corresponds to 100 effective passes; in all plots, a black dashed line marks the first effective pass), (4) compute averaged costs on training and testing data (based on 10 replications). All the costs are shown in log-scale, normalized to that the first iteration leads to $f(\theta_0) - f(\theta_*) = 1$.

All algorithms that we consider (ours and others) have a step-size, and typically a theoretical value that ensures convergence. We consider two settings: (1) one when this theoretical value is used, (2) one with the best testing error after one effective pass through the data (testing powers of $4$ times the theoretical step-size).

Here, we only consider *covertype*, *alpha*, *sido* and *news*, as well as test errors. For all training errors and the two other datasets (*quantum*, *rcv1*), see [15].

**Least-squares regression.** We compare three algorithms: averaged SGD with constant step-size, averaged SGD with step-size decaying as $C/R^2\sqrt{n}$, and the stochastic averaged gradient (SAG) method which is dedicated to finite training data sets [27], which has shown state-of-the-art performance in this set-up. We show the results in the two left plots of Figure 2 and Figure 3.

Averaged SGD with decaying step-size equal to $C/R^2\sqrt{n}$ is slowest (except for *sido*). In particular, when the best constant $C$ is used (right columns), the performance typically starts to increase significantly. With that step size, even after 100 passes, there is no sign of overfitting, even for the high-dimensional sparse datasets.

SAG and constant-step-size averaged SGD exhibit the best behavior, for the theoretical step-sizes *and* the best constants, with a significant advantage for constant-step-size SGD. The non-sparse datasets do not lead to overfitting, even close to the global optimum of the (unregularized) training objectives, while the sparse datasets do exhibit some overfitting after more than 10 passes.

**Logistic regression.** We also compare two additional algorithms: our Newton-based technique and "Adagrad" [7], which is a stochastic gradient method with a form a diagonal scaling[1] that allows to reduce the convergence rate (which is still in theory proportional to $O(1/\sqrt{n})$). We show results in the two right plots of Figure 2 and Figure 3.

Averaged SGD with decaying step-size proportional to $1/R^2\sqrt{n}$ has the same behavior than for least-squares (step-size harder to tune, always inferior performance except for *sido*).

SAG, constant-step-size SGD and the novel Newton technique tend to behave similarly (good with theoretical step-size, always among the best methods). They differ notably in some aspects: (1) SAG converges quicker for the training errors (shown in [15]) while it is a bit slower for the testing error, (2) in some instances, constant-step-size averaged SGD does underfit (*covertype*, *alpha*, *news*), which is consistent with the lack of convergence to the global optimum mentioned earlier, (3) the novel online Newton algorithm is consistently better.

On the non-sparse datasets, Adagrad performs similarly to the Newton-type method (often better in early iterations and worse later), except for the *alpha* dataset where the step-size is harder to tune (the best step-size tends to have early iterations that make the cost go up significantly). On sparse datasets like *rcv1*, the performance is essentially the same as Newton. On the *sido* data set, Adagrad (with fixed steps size, left column) achieves a good testing loss quickly then levels off, for reasons we cannot explain. On the *news* dataset, it is inferior without parameter-tuning and a bit better with. Adagrad uses a diagonal rescaling; it could be combined with our technique, early experiments show that it improves results but that it is more sensitive to the choice of step-size.

Overall, even with $d$ and $\kappa$ very large (where our bounds are vacuous), the performance of our algorithm still achieves the state of the art, while being more robust to the selection of the step-size: finer quantities likes degrees of freedom [13] should be able to quantify more accurately the quality of the new algorithms.

## 5 Conclusion

In this paper, we have presented two stochastic approximation algorithms that can achieve rates of $O(1/n)$ for logistic and least-squares regression, without strong-convexity assumptions. Our analysis reinforces the key role of averaging in obtaining fast rates, in particular with large step-sizes. Our work can naturally be extended in several ways: (a) an analysis of the algorithm that updates the support point of the quadratic approximation at every iteration, (b) proximal extensions (easy to implement, but potentially harder to analyze); (c) adaptive ways to find the constant-step-size; (d) step-sizes that depend on the iterates to increase robustness, like in normalized LMS [20], and (e) non-parametric analysis to improve our theoretical results for large values of $d$.

**Acknowledgements.** Francis Bach was partially supported by the European Research Council (SIERRA Project). We thank Aymeric Dieuleveut and Nicolas Flammarion for helpful discussions.

Table 1: Datasets used in our experiments. We report the proportion of non-zero entries, as well as estimates for the constant $\kappa$ and $\rho$ used in our theoretical results, together with the non-sharp constant which is typically used in analysis of logistic regression and which our analysis avoids (these are computed for non-sparse datasets only).

| Name | $d$ | $n$ | sparsity | $\kappa$ | $\rho$ | $1/\inf_n \ell''(y_n, \langle \theta_*, x_n \rangle)$ |
|------|-----|-----|----------|----------|--------|------------------------------------------------------|
| *quantum* | 79 | 50 000 | 100 % | $5.8 \times 10^2$ | 16 | $8.5 \times 10^2$ |
| *covertype* | 55 | 581 012 | 100 % | $9.6 \times 10^2$ | 160 | $3 \times 10^{12}$ |
| *alpha* | 501 | 500 000 | 100 % | 6 | 18 | $8 \times 10^4$ |
| *sido* | 4 933 | 12 678 | 10 % | $1.3 \times 10^4$ | $\times$ | $\times$ |
| *rcv1* | 47 237 | 20 242 | 0.2 % | $2 \times 10^4$ | $\times$ | $\times$ |
| *news* | 1 355 192 | 19 996 | 0.03 % | $2 \times 10^4$ | $\times$ | $\times$ |

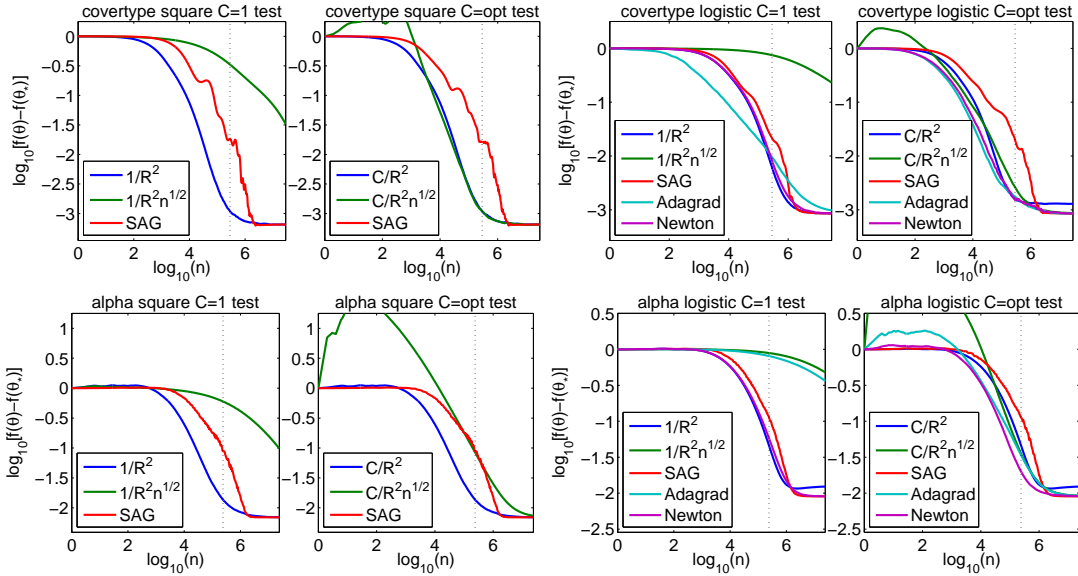

Figure 2: Test performance for least-square regression (two left plots) and logistic regression (two right plots). From top to bottom: *covertype*, *alpha*. Left: theoretical steps, right: steps optimized for performance after one effective pass through the data. Best seen in color.

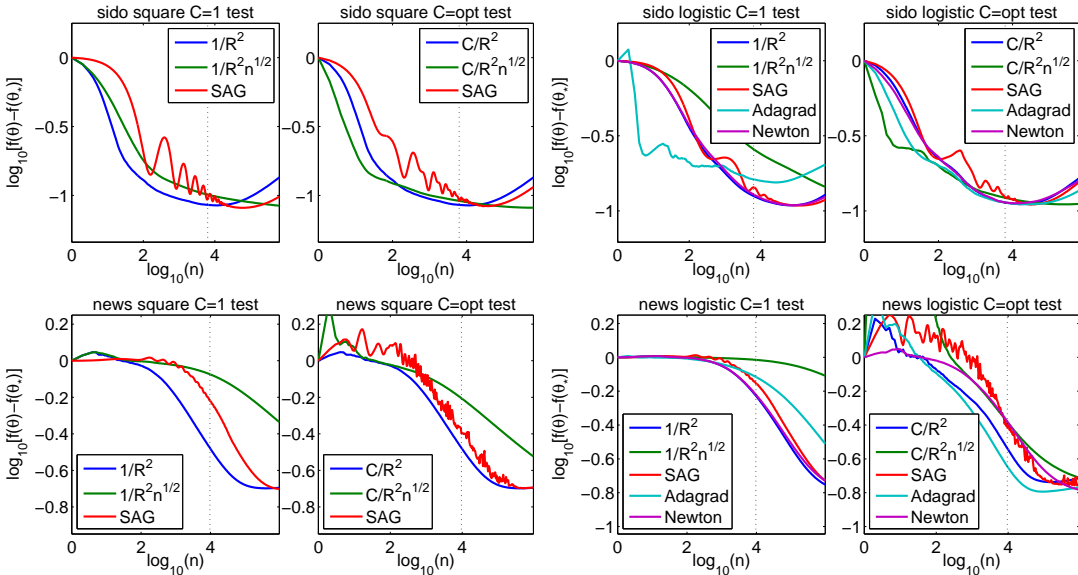

Figure 3: Test performance for least-square regression (two left plots) and logistic regression (two right plots). From top to bottom: *sido*, *news*. Left: theoretical steps, right: steps optimized for performance after one effective pass through the data. Best seen in color.

## Footnotes

[1]Since a bound on $\|\theta_*\|$ is not available, we have used step-sizes proportional to $1/\sup_n \|x_n\|_\infty$.

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
