[Reviews · NeurIPS 2013]

Submitted by Assigned_Reviewer_4

The paper presents novel results on how to achieve fast convergence rates O(1/n) in on-line learning via stochastic approximation with requiring strong convexity of the loss function and without having to rely on well-conditioned data matrices. This addresses one of the most interesting open questions in on-line learning, namely how to achieve convergence rates closer to the theoretical limit for the practically most relevant settings (least squares and logistic regression). In my opinion, the paper makes a number of remarkable contributions towards this goal: (i) Theorem 1: a fast convergence result for constant-step size least squares methods with averaging iterates, and an analysis of convergence of higher order moments and (ii) a clever way on how to extend this from the linear case to the more general case (of M-estimators) via the notion of "support points". While the fully satisfactory exploitation of the latter idea seems to require future work (the two phase method seems somewhat less appealing from a practical point of view and the the algorithm who choses support points on the fly is not known to converge), I feel this is already worth publishing in order to make these ideas accessible to a broader audience.
Summary: Great paper on fast convergence algorithms without requiring strong convexity. Some really novel and inspiring ideas. Promising experimental results.

Submitted by Assigned_Reviewer_5

Quality:
Good work.

Clarity:
The paper is well written. Some comments:
1. Does notation “x_n \otimes x_n” represent the Kronecker product? If so, should be “x_n \otimes x_n^T”? Why not directly use x_n x_n^T?
2.“{\cal H} is a d-dimensional Euclidean space.” Why not use notation {\mathbb R}^d?

Originality:
The work is novel.

Significance:
The results in Theorems 1, 2 and 3 are interesting and important. But I don’t fully check the proofs.

Summary: This paper focuses on optimization problems without strong convexity, for which most of the previous methods have a convergence rate of O(1/\sqrt{n}). This paper then developed two algorithms with convergence rate O(1/n), which are respectively for least-squares regression and logistic regression. Moreover, the paper conducted empirical analysis on synthetic data and standard benchmarks.

Submitted by Assigned_Reviewer_7

This paper shows that for some relevant machine learning problems involving the stochastic minimization of smooth convex functions, it is possible to get rates of 1/n even though the strong convexity parameter can be arbitrarily small. This claim seems important and relevant to the community if it was true.

This paper is not very well written even though it reflects technical expertise. Hardly any proofs are given in the main paper, and almost no intuition for the proofs is given too. It wasn't very clear why exactly the claimed result was possible. The claim seems to be important if true, but I struggled to verify how or why it was true.

Too many pages were spent on experiments including an entire page of graphs, which are unlikely to be convincing (or too much space spent on it) if the theorems and proofs have limited intuition. I would throw out some graphs but instead justify the theory better.
Summary: I think that the paper can be important and significant if it was correct - however the paper is not written clearly and with enough details for me to judge whether this was the case - however the experiments might suggest that the theory is correct and hence this may be of interest to the community for both theory and practice.
Author Feedback

Author rebuttal: We would like to thank the reviewers for their constructive comments. A few additional points:

( ) Significance/impact:
Our paper addresses with a simple algorithm parts of a recent open problem posed at COLT 2013:
https://orfe.princeton.edu/conferences/colt2013/sites/orfe.princeton.edu.conferences.colt2013/files/koren-open.pdf

This is done by introducing new concepts and proof techniques borrowed and adapted from the signal processing community.

( ) Proofs vs. experiments:
It is always a struggle in ML/optimization papers to decide between these. We kept the graphs to show that our simple algorithm (one line of code on top of SGD) had a better theoretical bound, O(1/n) instead of O(1/sqrt(n)), but also a good matching empirical behavior (fast convergence and robustness to step-size). If accepted, in the final version, we may remove one or two datasets to leave more space for prook sketches and expand the homogeneous Markov chain intuition (which provides a one-line proof why the rate is O(1/n)).

( ) Clarification on notations:
The paper was originally written for Hilbert spaces hence the Kronecker product, {\cal H} and the <,> notations. In order to keep it simple we have removed this aspect (which requires to define appropriately a notion of data-dependent dimension similar to degrees of freedom).